# Quality Evaluation of Rock Mass Using RMR14 Based on Multi-Source Data Fusion

**DOI:** 10.3390/s21217108

**Published:** 2021-10-26

**Authors:** Qi Zhang, Qing Jiang, Yuanhai Li, Ning Wang, Lei He

**Affiliations:** 1School of Civil Engineering, Southeast University, Nanjing 211189, China; jiangqing@seu.edu.cn (Q.J.); wangning97@seu.edu.cn (N.W.); helei_civil@seu.edu.cn (L.H.); 2State Key Laboratory for GeoMechanics and Deep Underground Engineering, China University of Mining & Technology, Xuzhou 221116, China; lyh@cumt.edu.cn; 3Institute of Future Underground Space, Southeast University, Nanjing 211189, China

**Keywords:** rock mass, quality evaluation, data-driven computing, multi-source data fusion, D-S evidence theory, belief reinforcement

## Abstract

The uncertainties in quality evaluations of rock mass are embedded in the underlying multi-source data composed by a variety of testing methods and some specialized sensors. To mitigate this issue, a proper method of data-driven computing for quality evaluation of rock mass based on the theory of multi-source data fusion is required. As the theory of multi-source data fusion, Dempster–Shafer (D-S) evidence theory is applied to the quality evaluation of rock mass. As the correlation between different rock mass indices is too large to be ignored, belief reinforcement and Murphy’s average belief theory are introduced to process the multi-source data of rock mass. The proposed method is designed based on RMR14, one of the most widely used quality-evaluating methods for rock mass in the world. To validate the proposed method, the data of rock mass is generated randomly to realize the data fusion based on the proposed method and the conventional D-S theory. The fusion results based on these two methods are compared. The result of the comparison shows the proposed method amplifies the distance between the possibilities at different ratings from 0.0666 to 0.5882, which makes the exact decision more accurate than the other. A case study is carried out in Daxiagu tunnel in China to prove the practical value of the proposed method. The result shows the rock mass rating of the studied section of the tunnel is in level III with the maximum possibility of 0.9838, which agrees with the geological survey report.

## Highlights

1. A complete method using RMR14 for quality evaluation of rock mass is proposed based on the Dempster–Shafer (D-S) evidence theory, which is used for multi-source data fusion.

2. Belief reinforcement is applied to process the rock mass data as evidence to reduce the impact brought by the correlation between rock mass indices.

3. Murphy’s Average Belief Theory is considered in the combination rule of D-S evidence theory to widen the gap between the possibilities of each rating level to reach more reasonable decisions.

## 1. Introduction

The quality evaluation of rock mass is fundamental to the survey, design, and construction of rock engineering projects. This kind of quality evaluation is mostly based on specific rating methods, including Rock quality designation (RQD) [1,2], the Geological strength index (GSI) [3,4], Rock mass rating (RMR89 and RMR14) [5,6,7], modified BQ [8], and so on. All these methods need rock mass indices before evaluating the quality of rock mass. Numerous testing methods to obtain these indices lead to the uncertainty of the evaluation result. RMR14 evaluates the quality by rating five indices of rock mass, which are rock strength, the spacing of discontinuities, the condition of discontinuities, ground water, and intact rock alterability [5]. There are multiple choices to obtain these indices. Rock strength can be obtained by the point load test and the Schmidt hammer test [9,10,11,12]. Both the spacing and the condition of discontinuities can be obtained by field observation, digital photography, and laser scanning [13,14,15]. The index of ground water can be obtained by field observation and ground-penetrating radar (GPR) scanning [16]. Intact rock alterability can be obtained by laboratory test [17] or be converted from the experimental result of the point load test through an empirical formula [18]. Since the RMR14 system is the most systematic evaluation method for the rock mass in tunnel engineering projects, the study evaluates the quality of rock mass based on this system by using multi-source data fusion.

The data of rock mass obtained by various methods can be defined as the multi-source data of rock mass. In order to evaluate the quality of rock mass better based on these multi-source data, a proper method based on the theory of multi-source data fusion, as a way of data-driven computing, is needed. Data-driven computing has been widely used in many fields, such as traction systems in high-speed trains [19]. Multi-source data fusion, as one of the data-driving computing methods, is a technology that automatically analyzes and processes target data and information from multiple sources, including sensors and experiments. It can draw conclusions or make decisions according to time series and criteria [20,21,22,23]. Multi-source data fusion is generally classified into three levels: data level, feature level, and decision level. The current studies about multi-source data fusion in engineering fields are mostly focused on the application of theoretical methods in actual engineering projects. Razavi and Haas used multi-sensor data fusion to track the materials automatically during the construction [24]. Wang et al. applied machine learning and data fusion technology to environmental sensing in buildings [25]. Tran et al. introduced a method based on multi-sensor data fusion for milling chatter detection, which is cheaper and easier compared with traditional chatter detection schemes [26]. Liu et al. proposed a decision-level sensor fusion based on the Sugeno fuzzy integral to integrate the vibration and current information for more accurate diagnosis [27]. Bakr and Lee review the theories of multi-sensor data fusion and highlight the directions of future research in multi-sensor data fusion under unknown correlation and data inconsistency [28].

The theories of multi-source data fusion include Bayesian theory [29], Dempster–Shafer (D-S) evidence theory [30,31], neural networks [32], and so on. Bayes theory was first proposed by British mathematician Thomas Bayes, which calculates the posterior probabilities based on the known prior knowledge and sample information. Bayes theory connects the prior knowledge to the posterior probabilities [29,33]. The D-S evidence theory is an uncertain reasoning theory proposed by Dempster and perfected by Shafer, which regards diverse data as evidence and fuses it based on its belief function [30,31,34]. Compared to Bayes theory, its demand for prior information is weaker. This theory can directly express the unknown and uncertainty, which can be retained in the process of fusion [35]. A neural network is a nonlinear self-adaptive and self-organizing system composed of many simple processing units which simulate the human nervous system to process information based on the research of biological neuroscience [32,36,37]. The neural network can fully approximate any complex nonlinear relationship and store information in neurons in the system with strong robustness and fault tolerance. With the self-learning and self-adapting ability, the neural network system can also deal with data fusion problem. Zhang et al. apply the neural network as a fusion method to predict the mechanical condition dynamically [38]. Compared to other two theories, neural network is more applicable for deep learning than data fusion [37].

The required prior data in D-S evidence theory is more intuitive and easier to obtain than a probabilistic reasoning theory such as Bayesian theory. It can integrate a variety of data and knowledge, which makes it easier for the further data fusion. The study applies D-S evidence theory as the theory of multi-source data fusion to fuse the multi-source data of rock mass. The level of data fusion is at the decision level. In D-S evidence theory, the most critical step is to determine the basic probability assignments (BPAs) of each evidence in the discernment framework for decision making. If the correlations between each evidence are not entirely considered, it would lead to inaccurate and unreliable fusion results [39,40,41,42,43]. To make the results of data fusion more convincing and accurate for decision making, the study needs to process the data of rock mass as evidence first before further fusion. Researchers have conducted some research for this purpose. Murphy’s average belief theory is proposed [40], which calculates the average belief of evidence for further fusion. Many other methods are introduced to calculate the BPAs and the belief of evidence in D-S theory for the same purpose, including the Jensen–Shannon belief divergence measurement to reinforce the belief of evidence [41,42,43] and the distance measurement between each evidence [43,44].

This study proposes a method for multi-source quality evaluation of rock mass based on D-S evidence theory. As the correlations between different properties of rock mass cannot be ignored, the belief reinforcement is applied to process the prior data of rock mass for the further fusion process. Murphy’s average belief theory is used in the further data fusion process for more reliable decision making, which is different from the applications of the traditional D-S evidence theory. To validate the proposed method, random data is generated, and the fusion results by both conventional D-S theory and the proposed method are compared. A practical case study is processed by the proposed method to prove its practical value.

## 2. D-S Evidence Theory

### 2.1. Description of D-S Evidence Theory

The basic concept of D-S evidence theory includes the discernment framework, a basic probability assignment, a belief function, and a likelihood function [28,29].

The discernment framework is a finite nonempty set Θ including all the possible events for the problem need to be decided. It can be expressed as $\Theta =\left\{\theta_{1},\theta_{2},\ldots ,\theta_{n}\right\}$.

The basic probability assignment (BPA) represents the trust degree of evidence for the possible events in Θ, according to the data obtained. The definition of BPA is as follows:

Let $2^{\Theta }$ denote the power set composed of all the possible subsets of Θ. BPA is a function *m* mapping from $2^{\Theta }$ to [0, 1], and satisfies the Equation (1):(1)$$\{\begin{array}{c} m(\emptyset )=0\\ \sum_{A\subseteq \Theta }m(A)=1 \end{array}$$
where *A* is a subset of Θ (*A* can also be called an event), and *m*(*A*) is called subset *A*’s BPA, representing the trust degree of *A* in the framework Θ according to the data as evidence. m(∅)=0 invalidates that there is no trust for the empty set. The total trust degree for all events in the framework should be 1.

For each event in Θ, the sum of its subset’s BPAs is called the belief function, written as *Bel*. The definition is as follows: *Bel* is a function mapping from $2^{\Theta }$ to [0, 1] and satisfies Equation (2):(2)$$Bel(A)=\sum_{B\subseteq A}m(B)$$
where *Bel*(*A*) is event *A*’s belief function, indicating the degree to trust the true proposition of event *A*.

The likelihood function indicates the suspicious degree to the false proposition of each event, which is for the more comprehensive analysis of the event. The definition is as follows: *Pl* is a function mapping from $2^{\Theta }$ to [0, 1] and satisfies Equation (3):(3)$$Pl(A)=1-Bel(\bar{A})$$
where *Pl*(*A*) is used as the likelihood function of event *A*, indicating the trust degree for event *A* not to be a false proposition. *Bel*(*A*) indicates the suspicious degree of the false proposition of event *A*.

The relationship between the belief function and the likelihood function can be visually represented as Figure 1, as well as the uncertain relationship between the data as evidence.

### 2.2. Combination Rule of D-S Theory

When there is more evidence to analyze and process in the same discernment framework, the combination rule of D-S evidence [30,34] is required. The BPAs of each evidence are combined to a total BPA through this combination rule, which has proven to satisfy the commutative and associative laws. There is no requirement to consider the sequence of evidence combinations [30]. This rule reduces the reliability of evidence with a large base and amplifies the reliability of evidence with a small base, which is conducive to making a more reliable decision.

The combination rule of D-S evidence theory is realized through the orthogonal sum operation of several BPAs. *m*_1_, *m*_2_, …, *m_n_* are BPAs in the same discernment framework are set. $m_{1}⊕m_{2}$ represents the combination between two evidences’ BPAs. The calculating process is indicated by Equation (4):(4)$$m_{1}⊕m_{1}⊕\ldots ⊕m_{n}(A)=\frac{1}{K}\sum_{A_{1}\cap \ldots \cap A_{n}=A}m_{1}(A_{1})m_{2}(A_{2})\ldots m_{n}(A_{n})$$
where *K* is the conflict coefficient and calculated by the Equation (5):(5)$$K=\sum_{A_{1}\cap \ldots \cap A_{n}\neq \phi }m_{1}(A_{1})m_{2}(A_{2})\ldots m_{n}(A_{n})=1-\sum_{A_{1}\cap \ldots \cap A_{n}=\phi }m_{1}(A_{1})m_{2}(A_{2})\ldots m_{n}(A_{n})$$

### 2.3. Measurement for Basic Probability Assignment

D-S evidence theory has constructed a strict theoretical system and proved a feasible method to combine the BPAs of evidence. A reasonable method to obtain the BPAs plays a critical role in the application of D-S evidence theory [30,31]. Considering that the RMR14 has divided the quality of rock mass into five rating levels which have their own rating ranges, the Euclidian distance formula [45] can be used to transform the rock mass data into BPAs.

If the number of indices in a sample is *p*, which represents the different properties of this sample, this sample can be regarded as a *p*-dimensional vector. There are *n* similar samples, which form *p* dimensions. The difference between any two samples can be referred to as the distance between them in the broad sense, and the Euclidian distance formula is used for the calculation of such distance as in Equation (6):(6)$$d_{ij}=\sqrt{\sum_{k=1}^{p}{\left(x_{ik}-x_{jk}\right)}^{2}}(i,j=1,2,\ldots ,n)$$
where *d_i_*_j_ means the distance between sample *i* and *j*; the expressions of these two samples are $X_{i}=\{x_{i1},x_{i2},\ldots ,x_{ip}\}$ and $X_{j}=\{x_{j1},x_{j2},\ldots ,x_{jp}\}$, respectively.

In D-S evidence theory, it can be assumed that there is a discernment framework $\Theta =\{X_{1},X_{2},\ldots ,X_{n}\}$. *X_i_* (*i* = 1, 2, …, *n*) in the framework is a target pattern and can be written as $X_{i}=\{x_{i1},x_{i2},\ldots ,x_{ip}\}$. Each evidence can be written as $S_{i}=\{s_{i1},s_{i2},\ldots ,s_{ip}\}$, then the distance *D_ij_* between each evidence to the target pattern in the framework can be calculated as the Equation (7):(7)$$D_{ij}=\sqrt{\sum_{k=1}^{p}{\left(s_{ik}-x_{jk}\right)}^{2}}(i,j=1,2,\ldots ,n)$$

Then, the BPA of the evidence can be generated by Equation (8):(8)$$m_{s_{i}}(X_{j})=\frac{1/D_{ij}}{\sum_{k=1}^{n}\left(1/D_{ik}\right)}$$
where $m_{s_{i}}(X_{j})$ represents the BPA of evidence *S_i_* to the target pattern *X_j_*.

### 2.4. Measurement for Belief Reinforcement

Conventional D-S evidence theory supposes the evidence be independent to each other. Evidence could hardly meet this assumption in real work, which would influence badly on the reliability of the result. Research has been done to consider the correlations between evidence to reduce such impact. Murphy [40] suggested incorporating average belief into the combining rule, which is called Murphy’s average belief theory. Deng et al. [46] used the distance measurement to take the weight of each evidence into account based on Murphy’s average belief theory. Divergence and information entropy are introduced into the fusion process of D-S evidence theory by Xiao [42] to reinforce the belief of BPAs for the weight calculation. The study applied Xiao’s method [42,43] to complete the belief reinforcement of BPAs to reduce the influence brought by the correlation between data of rock mass as evidence.

To apply the measurement for belief reinforcement, the divergence measurement for belief functions is needed first as follows:

It is supposed that the discernment framework be $\Theta =\{X_{1},X_{2},\ldots ,X_{n}\}$ with belief functions *m*_1_, *m*_2_, …, *m_n_*. The belief divergence measurement, denoted by *B*, between any two belief functions *m_i_* and *m_j_* can be defined as:(9)$$B(m_{i},m_{j})=\sum_{p=1}^{2^{\Theta }}\sum_{q=1}^{2^{\Theta }}m_{i}(X_{p})\ln\frac{m_{i}(X_{p})}{\frac{1}{2}m_{i}(X_{p})+\frac{1}{2}m_{j}(X_{q})}\frac{\left|X_{p}\cap X_{q}\right|}{\left|X_{q}\right|}+\sum_{p=1}^{2^{\Theta }}\sum_{q=1}^{2^{\Theta }}m_{j}(X_{p})\ln\frac{m_{j}(X_{q})}{\frac{1}{2}m_{i}(X_{p})+\frac{1}{2}m_{j}(X_{q})}\frac{\left|X_{p}\cap X_{q}\right|}{\left|X_{p}\right|}$$
where $X_{p}\cap X_{q}$ means the intersection between *X_p_* and *X_q_*, and $\left|X_{p}\right|$ means the cardinality of *X_p_*.

When the events of the belief functions are one-element sets, Equation (9) degenerates into the belief Jensen–Shannon divergence as in Equation (10):(10)$$B(m_{i},\;m_{j})=\sum_{p=1}^{2^{\Theta }}m_{i}(X_{p})\ln\frac{m_{i}(X_{p})}{\frac{1}{2}m_{i}(X_{p})+\frac{1}{2}m_{j}(X_{p})}+\sum_{p=1}^{2^{\Theta }}m_{j}(X_{p})\ln\frac{m_{j}(X_{p})}{\frac{1}{2}m_{i}(X_{p})+\frac{1}{2}m_{j}(X_{p})}$$

Based on the divergence measurement for belief functions, the measurement for belief reinforcement, denoted by *RB*, between any two belief functions *m_i_* and *m_j_* can be defined as Equation (11). $RB(m_{i},\;m_{j})$ is the result of belief reinforcement for the further data fusion:(11)$$RB(m_{i},\;m_{j})=\sqrt{\frac{\left|B(m_{i},\;m_{i})+B(m_{j},\;m_{j})-B(m_{i},\;m_{j})-B(m_{j},\;m_{i})\right|}{2}}$$

It has been proved that both divergence measurement and measurement for belief reinforcement are central symmetry, which means $B(m_{i},\;m_{j})=B(m_{j},\;m_{i})$ and $RB(m_{i},\;m_{j})=RB(m_{j},\;m_{i})$.

## 3. A Method for Quality Evaluation of Rock Mass with Multi-Source Data

### 3.1. Description

The proposed method applies D-S evidence theory with the measurement for belief reinforcement to fuse the multi-source data of rock mass for the quality evaluation of rock mass in RMR14. The data of rock mass, including rock strength, spacing of discontinuities, condition of discontinuities, ground water, and intact rock alterability (*I_d_*_2_), are obtained by several test approaches. Some of them are collected by the installed sensors in tunnel excavation face, and others are measured by rock experiments. These data are converted into BPAs by the Euclidian distance formula. The data of rock mass are regarded as evidence in this method. Considering the correlations between the properties of rock mass, the measurement for belief reinforcement is applied to adjust the BPAs and calculate the weighted average belief of each evidence. The weighted average belief is iteratively fused by the combination rule of D-S evidence theory to obtain the final result. The result indicates the probability of the rock mass in each level of rock mass rating.

### 3.2. Process Steps

**Step 1:** According to the rock mass rating from level I to level V, the discernment framework as $\Theta =\left\{F_{I},\;F_{\mathrm{II}},\;F_{\mathrm{III}},\;F_{\mathrm{IV}},\;F_{V}\right\}$ is constructed, where *F_i_* (*i* = I, II, III, IV, V) means the quality of rock mass is in level *i* (*i* = I, II, III, IV, V).

**Step 2:** Rating ranges for each rock mass rating index are standardized, which is listed in Table 1.

The rock mass rating index *R*_3_ can be divided into four sub-indicators, as continuity *R*_31_, roughness *R*_32_, infilling *R*_33_, and weathering *R*_34_. Secondary data fusion is needed in *R*_3_ before the data fusion in the whole quality evaluation of rock mass. The rating ranges of the sub-indicators of *R*_3_ are listed as Table 2.

**Step 3:** The data of rock mass need to be transformed into BPAs by the Euclidian distance formula. The rock mass data can be obtained from various approaches, such as point load rest and Schmidt hammer test to obtain the data of rock strength, digital photography, and laser scanning to obtain data of rock discontinuities, field observation, and GPR scanning to obtain data of ground water. For each rock mass data, it has a corresponding rating *q* and can be written as $\left[q^{-},\;q^{+}\right]$, where *q*^−^ = *q*^+^ = *q*. The standardized rating ranges can also be written as $\left[f_{i}^{-},\;f_{i}^{+}]\;(i=I,\;\mathrm{II},\;\mathrm{III},\;\mathrm{IV},\;V\right)$, where $f_{i}^{-}$ represents the minimum rating of the range, and the $f_{i}^{+}$ represents the maximum rating of the range. The distance from the data to the rating range can be derived as Equation (12):(12)$$D_{i}=\sqrt{{\left(q^{-}-f_{i}^{-}\right)}^{2}+{\left(q^{+}-f_{i}^{+}\right)}^{2}}\;(i=I,\;\mathrm{II},\;\mathrm{III},\;\mathrm{IV},\;V)$$

Then, the BPA of this data of rock mass for each interval can be calculated as Equation (13):(13)$$m(F_{i})=\frac{1/D_{i}}{\sum_{k=1}^{5}\left(1/D_{k}\right)}(i=I,\;\mathrm{II},\;\mathrm{III},\;\mathrm{IV},\;V)$$

**Step 4:** *BM* is constructed as the BPA matrix according to the result from Step 3 as Equation (14):(14)$$BM=\left[\begin{array}{ccccc} m_{1}(F_{I}) & m_{1}(F_{\mathrm{II}}) & m_{1}(F_{\mathrm{III}}) & m_{1}(F_{\mathrm{IV}}) & m_{1}(F_{V})\\ m_{2}(F_{I}) & m_{2}(F_{\mathrm{II}}) & m_{2}(F_{\mathrm{III}}) & m_{2}(F_{\mathrm{IV}}) & m_{2}(F_{V})\\ \ldots & \ldots & \ldots & \ldots & \ldots \\ m_{n}(F_{I}) & m_{n}(F_{\mathrm{II}}) & m_{n}(F_{\mathrm{III}}) & m_{n}(F_{\mathrm{IV}}) & m_{n}(F_{V}) \end{array}\right]$$
where $m_{i}(F_{j})\;(i=1,2,\ldots ,\;n;\;j=I,\;\mathrm{II},\;\mathrm{III},\;\mathrm{IV},\;V)$ means the BPA for the data of rock mass to one rock mass rating range. The data of rock mass is obtained from one test method.

The divergence measurement for belief reinforcement can be derived as Equations (15) and (16), for the events in the framework are all one-element sets:(15)$$\begin{array}{ll} B(m_{i},m_{j})= & \sum_{p=I,\;\mathrm{II},\;\mathrm{III},\;\mathrm{IV},\;V}m_{i}(F_{p})\ln\frac{m_{i}(F_{p})}{\frac{1}{2}m_{i}(F_{p})+\frac{1}{2}m_{j}(F_{p})}\\ & +\sum_{p=I,\;\mathrm{II},\;\mathrm{III},\;\mathrm{IV},\;V}m_{j}(F_{p})\ln\frac{m_{j}(F_{p})}{\frac{1}{2}m_{i}(F_{p})+\frac{1}{2}m_{j}(F_{p})}\;(i,j=1,2,\ldots ,n) \end{array}$$
(16)$$RB_{ij}=RB(m_{i},m_{j})=\sqrt{\frac{\left|B(m_{i},m_{i})+B(m_{j},m_{j})-B(m_{i},m_{j})-B(m_{j},m_{i})\right|}{2}}\;(i,j=1,2,\ldots ,n)$$

The calculation result by Equation (16) is used to construct *RBM* as the matrix of belief reinforcement as Equation (17):(17)$$RBM=\left[\begin{array}{ccccc} RB_{11} & RB_{12} & RB_{13} & \ldots & RB_{1n}\\ RB_{21} & RB_{22} & RB_{23} & \ldots & RB_{2n}\\ \ldots & \ldots & \ldots & \ldots & \ldots \\ RB_{n1} & RB_{n2} & RB_{n3} & \ldots & RB_{nn} \end{array}\right]$$

**Step 5:** The elements in *RBM* are used to calculate the weight average belief of each evidence as Equations (18)–(21):(18)$${\tilde{RB}}_{i}=\frac{\sum_{j=1}^{n}RB_{ij}}{n}$$
(19)$$Se_{i}=\frac{1}{{\tilde{RB}}_{i}}$$
(20)$$c_{i}=\frac{Se_{i}}{\sum_{j=1}^{n}Se_{j}}$$
(21)$$\tilde{m}(F_{j})=\sum_{i=1}^{5}\left[c_{i}\times m_{i}(F_{j})\right]\;(j=I,\;\mathrm{II},\;\mathrm{III},\;\mathrm{IV},\;V)$$

**Step 6:** The result from step 5 is used to build a weighted average belief vector as Equation (22):(22)$$\tilde{m}=(\tilde{m}(F_{I}),\;\tilde{m}(F_{\mathrm{II}}),\;\tilde{m}(F_{\mathrm{III}}),\;\tilde{m}(F_{\mathrm{IV}}),\;\tilde{m}(F_{V}))$$

The combination rule of D-S evidence theory and Murphy’s average belief theory are applied to obtain the final fusion result as Equation (23):(23)$$\tilde{F}[\tilde{m}(F_{j})]={\left\{{\left\{{\left\{{\left[\tilde{m}(F_{j})⊕\tilde{m}(F_{j})\right]}_{1}⊕\tilde{m}(F_{j})\right\}}_{2}⊕\ldots \right\}}_{n-2}⊕\tilde{m}(F_{j})\right\}}_{n-1}$$

The flowchart of the proposed method is shown in Figure 2.

### 3.3. Model Validation

In order to validate the proposed method, 25 sets of data for rock mass rating are listed in Table 3. Generated data of rock mass indicate that this kind of rock mass is probably in level II or III. Each index (*R*_1_–*R*_5_) has 5 datasets used as evidence for data fusion. The BPAs of each evidence are calculated shown in Figure 3, which are reinforced by the measurement in Section 2.4 and shown as *RBM* in Equation (24). Results of the fusion process are listed in Table 4. Then the weighted average belief is calculated based on the proposed method and the BPAs, which is shown in Figure 4.
(24)$$RBM=\left[\begin{array}{ccccc} 0 & & & & \\ 0.6114 & 0 & & \mathrm{Symmetry} & \\ 0.4829 & 0.3941 & 0 & & \\ 0.3227 & 0.4558 & 0.1839 & 0 & \\ 0.3397 & 0.4267 & 0.1588 & 0.0353 & 0 \end{array}\right]$$

A further fusion step is performed as mentioned in Section 3.2. The fusion result is shown in Figure 5. The fusion result obtained by conventional D-S theory’s combination rule is also shown in Figure 5. It is found that the result obtained from conventional D-S theory shows almost the close probabilities in level II and level III, making it difficult to decide the exact level of rock mass rating. While the result obtained by the proposed method indicates that the actual rock mass rating level is III with the maximum probability of 0.8887. The weighted average belief also shows that is same as the result of the proposed method, which is magnified from 0.3005 to 0.8887 by the Murphy’s average belief theory. The simulation result indicates that the proposed method can process the multi-source data with correlations between each evidence in D-S theory better than the conventional theory and help to ensure more reliable decisions.

Meanwhile, the traditional way for the quality evaluations is also applied. The quality of rock mass in this simulation is evaluated as an exact value, **59.2**, which means the quality of this kind of rock mass is in level **III**. The result agrees with the proposed method, but the output expressions of the two methods are different. The proposed method, taking probability as the output mode, reflects the uncertainty in the quality evaluation of rock mass, which is more reliable in tunnel designation than the numerical output mode.

## 4. Case Study

### 4.1. Background

The proposed method is applied to the quality evaluation of rock mass in the Daxiagu tunnel under construction in the E’han expressway in the Sichuan province of China as a case study. The tunnel is the deepest buried highway tunnel with the maximum buried depth of 1944 m. The surrounding rock data used in this case study are collected from the K78+350 section of the tunnel in Figure 6. The rock data of rock mass rating indices are collected by several methods and some specialized sensors, including the point load test, the Schmidt hammer test, digital photography, laser scanning, field observation, GPR scanning, and so on. Some of them use sensors to obtain the data of rock mass, such as field observation. The quality evaluation of rock mass is generated by the proposed multi-source data fusion method.

### 4.2. Quality Evaluation with the Proposed Method

According to the proposed method in Section 3, the quality evaluation begins with initial data fusion in each rock mass index (*R*_1_–*R*_5_).

The rock mass index *R*_1_ is rated by rock strength, which is obtained by the point load test or the Schmidt hammer test. *R*_1_ is converted into BPAs, and $BM_{R_{1}}$ is constructed with BPAs of *R*_1_ as shown in Figure 7. $RBM_{R_{1}}$ can be calculated and constructed by Equation (25). The initial fusion of *R*_1_ for the proposed method is conducted and the result is shown in Figure 8a. It indicates that the quality of rock strength in this section has almost the same probability in level III (0.4981) and IV (0.5019). The probabilities of the quality of rock strength in level I, II, and V are almost 0. It is difficult to determine the exact quality of the strength of the rock mass in this section.
(25)$$RBM_{R_{1}}=\left[\begin{array}{cccccccccccccccc} 0 & & & & & & & & & & & & & & & \\ 0.2079 & 0 & & & & & & & & & & & & & & \\ 0.1902 & 0.0180 & 0 & & & & & & & & & & & & & \\ 0.2762 & 0.0694 & 0.0874 & 0 & & & & & & & & & & & & \\ 0.1212 & 0.0878 & 0.0698 & 0.1569 & 0 & & & & & & & \mathrm{Symmetry} & & & & \\ 0.1551 & 0.0535 & 0.0356 & 0.1228 & 0.0343 & 0 & & & & & & & & & & \\ 0.0742 & 0.1351 & 0.1172 & 0.2040 & 0.0475 & 0.0817 & 0 & & & & & & & & & \\ 0.0598 & 0.1494 & 0.1316 & 0.2182 & 0.0619 & 0.0962 & 0.0145 & 0 & & & & & & & & \\ 0 & 0.2079 & 0.1902 & 0.2762 & 0.1212 & 0.1551 & 0.0742 & 0.0598 & 0 & & & & & & & \\ 0.1049 & 0.1042 & 0.0862 & 0.1733 & 0.0164 & 0.0507 & 0.0311 & 0.0455 & 0.1049 & 0 & & & & & & \\ 0.2079 & 0 & 0.0180 & 0.0694 & 0.0878 & 0.0535 & 0.1351 & 0.1494 & 0.2079 & 0.1042 & 0 & & & & & \\ 0.2430 & 0.0356 & 0.0536 & 0.0339 & 0.1233 & 0.0891 & 0.1704 & 0.1848 & 0.2430 & 0.1396 & 0.0356 & 0 & & & & \\ 0.1379 & 0.0709 & 0.0529 & 0.1401 & 0.0169 & 0.0174 & 0.0644 & 0.0788 & 0.1379 & 0.0334 & 0.0709 & 0.1064 & 0 & & & \\ 0.1212 & 0.0878 & 0.0698 & 0.1569 & 0 & 0.0343 & 0.0475 & 0.0619 & 0.1212 & 0.0164 & 0.0878 & 0.1233 & 0.0169 & 0 & & \\ 0.0216 & 0.1873 & 0.1685 & 0.2558 & 0.1002 & 0.1343 & 0.0529 & 0.0385 & 0.0216 & 0.0838 & 0.1873 & 0.2225 & 0.1170 & 0.1002 & 0 & \\ 0.0372 & 0.2388 & 0.2213 & 0.3063 & 0.1537 & 0.1868 & 0.1081 & 0.0944 & 0.0372 & 0.1378 & 0.2388 & 0.2734 & 0.1700 & 0.1537 & 0.0579 & 0 \end{array}\right]$$

Rock mass index *R*_2_ is rated by the spacing of discontinuities, which is obtained by digital photography, and laser scanning. The excavation face is divided into several sampling windows, and the discontinuity information is extracted from the sampling windows to obtain the required data. The spacing of discontinuities is based on the number of discontinuities per unit length. $BM_{R_{2}}$ is constructed based on the BPAs of *R*_2_ in Figure 9. $RBM_{R_{2}}$ is calculated by Equation (26). The initial fusion of *R*_2_ for the proposed method is done and the result is shown in Figure 8b. From the result of initial fusion of *R*_2_, it indicates that the quality of the spacing of discontinuities in this section is in level II with the maximum probability of 0.8543. The result is different from the result of initial data fusion of *R*_1_, which means that further data fusion is needed.
(26)$$RBM_{R_{2}}=\left[\begin{array}{cccccccccccc} 0 & & & & & & & & & & & \\ 0.0500 & 0 & & & & & & & & & & \\ 0.3188 & 0.3646 & 0 & & & & & & & & & \\ 0.1766 & 0.2257 & 0.1644 & 0 & & & & & \mathrm{Symmetry} & & & \\ 0.1237 & 0.1730 & 0.2080 & 0.0533 & 0 & & & & & & & \\ 0.0628 & 0.1125 & 0.2615 & 0.1142 & 0.0610 & 0 & & & & & & \\ 0 & 0.0500 & 0.3188 & 0.1766 & 0.1237 & 0.0628 & 0 & & & & & \\ 0.0847 & 0.0373 & 0.3928 & 0.2570 & 0.2048 & 0.1452 & 0.0847 & 0 & & & & \\ 0.0628 & 0.1125 & 0.2615 & 0.1142 & 0.0610 & 0 & 0.0628 & 0.1452 & 0 & & & \\ 0.3047 & 0.3518 & 0.0355 & 0.1404 & 0.1883 & 0.2452 & 0.3047 & 0.3811 & 0.2452 & 0 & & \\ 0.0500 & 0 & 0.3646 & 0.2257 & 0.1730 & 0.1125 & 0.0500 & 0.0373 & 0.1125 & 0.3518 & 0 & \\ 0.2284 & 0.2772 & 0.1252 & 0.0525 & 0.1056 & 0.1663 & 0.2284 & 0.3080 & 0.1663 & 0.0956 & 0.2772 & 0 \end{array}\right]$$

Rock mass index *R*_3_ is rated by the condition of discontinuities. The sub-indicators of *R*_3_ are considered, including the rating of continuity, the rating of roughness, the rating of infilling, and the rating of weathering. The data of discontinuities of the rock mass are collected and rated, which is used to construct $BM_{R_{3}}$ in Figure 10. The continuity and the infilling of the rock mass is measured by digital photography and laser scanning based on the point load model [47]. The roughness of the rock mass is obtained by field observation and rock roughness coefficient based on rock mechanics [48]. The weathering condition of the rock mass is obtained by the field observation and rock tablet method [49]. The secondary fusion of each sub-indicator is carried out firstly before the initial fusion, as shown in Figure 11. The initial fusion of *R*_3_ based on the secondary fusion is conducted, and the result is shown in Figure 8c. It indicates that the quality of the condition of discontinuities in this section is in level III with the maximum probability of 0.9243. The fusion result still cannot reach a unified result with the fusion result of the previous two indicators.

Rock mass rating index *R*_4_ is rated by the ground water, which is obtained by field observation or GPR scanning. The index *R*_4_ is determined according to the wetting condition of ground water in the excavation face. The ground water at K78+350 was detected by both sources and the rock mass rating index *R*_4_ is rated. $BM_{R_{4}}$ is constructed with the rating as shown in Figure 12. $RBM_{R_{4}}$ is calculated by Equation (27). The initial fusion of *R*_4_ for the proposed method is performed, and the result is shown in Figure 8d. It indicates that the quality of ground water in this section has almost the same probability in level III (0.4761) and IV (0.4967), which makes it hard to decide the exact rating level of the quality of the ground water.
(27)$$RBM_{R_{4}}=\left[\begin{array}{cccccc} 0 & & & & & \\ 0.2102 & 0 & & \mathrm{Symmetry} & & \\ 0.2670 & 0.0581 & 0 & & & \\ 0.2336 & 0.0239 & 0.0342 & 0 & & \\ 0.1499 & 0.0617 & 0.1196 & 0.0855 & 0 & \\ 0.2878 & 0.0793 & 0.0213 & 0.0555 & 0.1407 & 0 \end{array}\right]$$

The rock mass rating index *R*_5_ is rated by the intact rock alterability (*I_d_*_2_). The standard test of this index is obtained by measuring the ratio of the residual mass of the rock specimen to its original mass after two standard cycles of drying and soaking. It is a long test period and is difficult to obtain [50]. According to the existing research, the rock intact rock alterability can be converted from the UCS obtained from point load test or laboratory test. The intact rock alterability index *R*_5_ of rock at the K78+350 section is rated and converted into BPAs in Figure 13. $RBM_{R_{5}}$ is constructed based on the BPAs in Equation (28). The initial fusion of *R*_5_ for the proposed method is done and the result is shown in Figure 8e, which indicates the quality of the rock intact rock alterability in this section is in level III with the maximum probability of 0.9558.
(28)$$RBM_{R_{5}}=\left[\begin{array}{cccccccccccccccc} 0 & & & & & & & & & & & & & & & \\ 0.4572 & 0 & & & & & & & & & & & & & & \\ 0.6532 & 0.3615 & 0 & & & & & & & & & & & & & \\ 0.5167 & 0.1081 & 0.2591 & 0 & & & & & & & & & & & & \\ 0.1004 & 0.5127 & 0.6887 & 0.5656 & 0 & & & & & & & \mathrm{Symmetry} & & & & \\ 0.3816 & 0.0940 & 0.4223 & 0.1890 & 0.4470 & 0 & & & & & & & & & & \\ 0.2292 & 0.5957 & 0.7436 & 0.6396 & 0.1295 & 0.5421 & 0 & & & & & & & & & \\ 0.2232 & 0.3169 & 0.5586 & 0.3888 & 0.3178 & 0.2275 & 0.4401 & 0 & & & & & & & & \\ 0.3159 & 0.1768 & 0.4687 & 0.2603 & 0.3922 & 0.0848 & 0.4989 & 0.1435 & 0 & & & & & & & \\ 0.4385 & 0.0248 & 0.3802 & 0.1305 & 0.4963 & 0.0701 & 0.5821 & 0.2951 & 0.1539 & 0 & & & & & & \\ 0.6230 & 0.3120 & 0.0532 & 0.2085 & 0.6611 & 0.3752 & 0.7201 & 0.5218 & 0.4251 & 0.3311 & 0 & & & & & \\ 0.5317 & 0.1403 & 0.2266 & 0.0331 & 0.5789 & 0.2171 & 0.6507 & 0.4077 & 0.2843 & 0.1619 & 0.1758 & 0 & & & & \\ 0.4481 & 0.0123 & 0.3711 & 0.1194 & 0.5047 & 0.0823 & 0.5891 & 0.3064 & 0.1657 & 0.0125 & 0.3218 & 0.1512 & 0 & & & \\ 0.5096 & 0.0928 & 0.2744 & 0.0157 & 0.5592 & 0.1758 & 0.6343 & 0.3799 & 0.2491 & 0.1156 & 0.2240 & 0.0487 & 0.1043 & 0 & & \\ 0.5567 & 0.1918 & 0.1746 & 0.0858 & 0.6012 & 0.2632 & 0.6694 & 0.4393 & 0.3242 & 0.2124 & 0.1233 & 0.0528 & 0.2022 & 0.1015 & 0 & \\ 0.4736 & 0.0247 & 0.3404 & 0.0846 & 0.5273 & 0.1167 & 0.6078 & 0.3362 & 0.1977 & 0.0492 & 0.2905 & 0.1172 & 0.0369 & 0.0691 & 0.1693 & 0 \end{array}\right]$$

The initial data fusion in each rock mass index is generated by the proposed method before the further fusion. The result of initial fusion of each index is shown in total in Figure 14. Some rock mass rating indices have different rating levels with the maximum probability (*R*_1_, *R*_3_, and *R*_5_), and others have almost the same probability in two rating levels (*R*_2_ and *R*_4_). It is difficult to decide the exact rating level of the quality of the rock mass in this section, which means that further fusion is needed for the quality evaluation of rock mass.

The result of the initial fusion of each index for the rock mass rating is used to conduct further data fusion. The total *RBM* of all the rock mass indices is calculated based on the results of initial data fusion by Equation (29). The further fusion process in Table 5 indicates that the rock mass rating of the K78+350 section is in level III with the maximum probability of 0.9838, which agrees with geological survey report of the Daxiagu tunnel. Equation (29) is calculated as follows:(29)$$RBM=\left[\begin{array}{ccccc} 0 & & & & \\ 1.0206 & 0 & & \mathrm{Symmetry} & \\ 0.6754 & 0.8374 & 0 & & \\ 0.1382 & 0.9909 & 0.6661 & 0 & \\ 0.5755 & 0.9460 & 0.2103 & 0.5805 & 0 \end{array}\right]$$

## 5. Conclusions

A multi-source quality evaluation of rock mass based on D-S evidence theory is proposed. Multi-source data of rock mass from different rock experiments or installed sensors can be entirely used in this method. The proposed method considers the correlation between different rock mass indices with measurement for belief reinforcement and applies the Murphy’s average belief theory to the combination rule of D-S evidence theory. The result obtained by the proposed method further enlarges the probability of more likely events and reduces the probability of less likely events.

Comparing this with the result based on the conventional combination rule of D-S theory, the proposed method processes the data of rock mass before further data fusion instantly. The results of the proposed method are also more accurate and reliable than the conventional D-S evidence theory. Compared to the traditional method based on RMR14 system, the output mode as probabilities shows the uncertainty in the quality evaluation of rock mass and makes the proposed method more reliable to the tunnel designation.

The K78+350 section of the Daxiagu tunnel is chosen as a case study to apply the proposed method. The result indicates that the quality of the rock mass is in level III. It agrees well with the reality and the geological survey report. The proposed method considers the correlation between rock properties together with the error influence caused by the accuracy of sensors and the operational problems. Its output is the probabilities in each rating level instead of the actual rating score, which reminds us to consider the risks caused by the uncertainty of tunnel surrounding rock in actual engineering projects. The proposed method can be taken forward for the application in the quality evaluation of rock mass with multi-source data in geotechnical engineering.

It has not been fully understood the mutual influence of the rating indices of the rock mass. For further research work, the interactions between the rating indices should be considered to address this limitation. A comprehensive approach is needed to process the correlations of the rating indices as evidence of the data fusion. Moreover, the RMR14 system was improved to evaluate the quality of the rock mass surrounding the tunnel in tunnel engineering. So, the proposed method is implemented in a case study of rock mass tunnel. The applicability of the proposed method in other rock mass engineering projects such as the side slope or underground cavern needs more studies in the future.

## Figures and Tables

**Figure 1 sensors-21-07108-f001:**
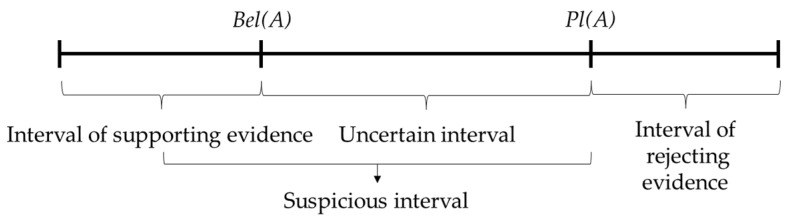
Uncertain relationship in D-S evidence theory.

**Figure 2 sensors-21-07108-f002:**
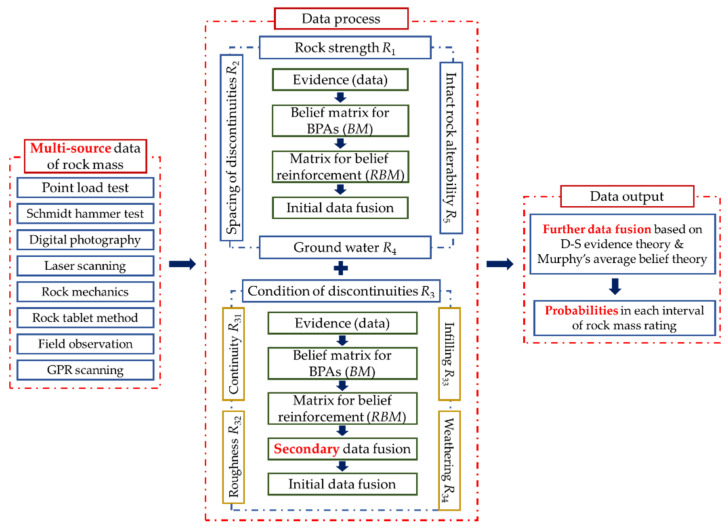
Flowchart of the proposed method.

**Figure 3 sensors-21-07108-f003:**
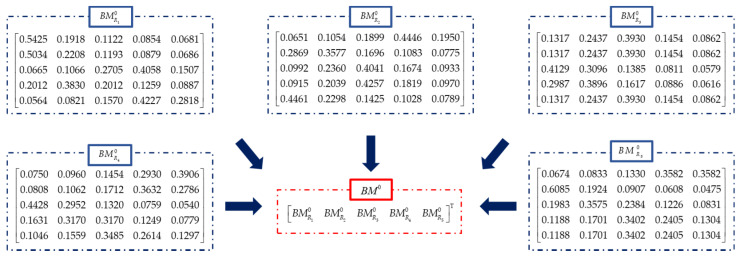
BPAs of each dataset as evidence.

**Figure 4 sensors-21-07108-f004:**
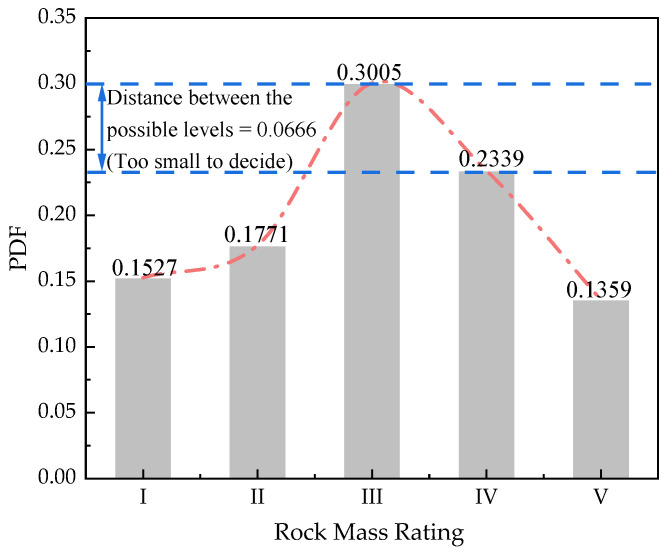
Weighted average belief of evidence in each interval of rock mass rating.

**Figure 5 sensors-21-07108-f005:**
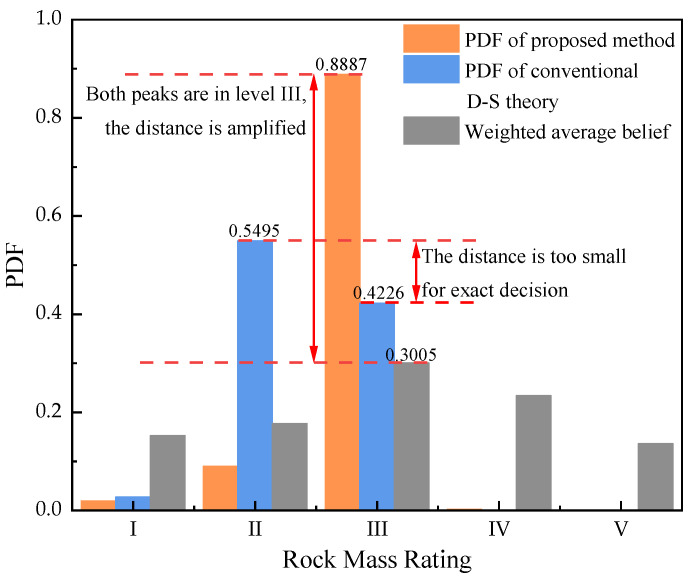
Comparison of fusion results.

**Figure 6 sensors-21-07108-f006:**
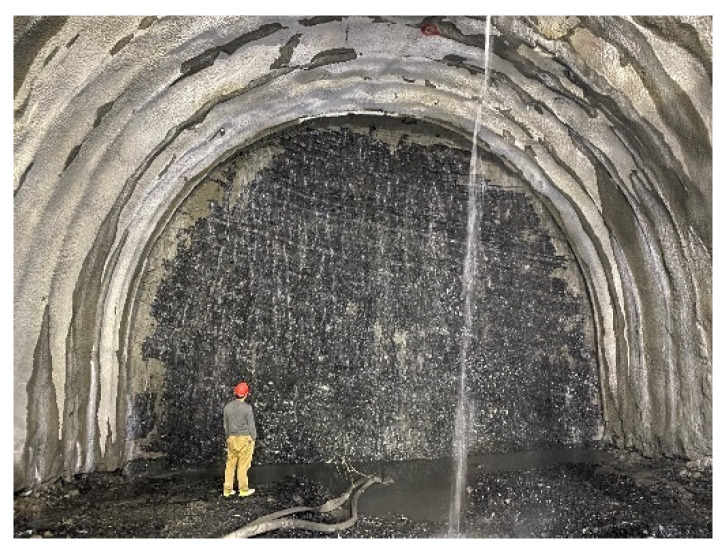
Tunnel excavation face at the K78+350 section.

**Figure 7 sensors-21-07108-f007:**
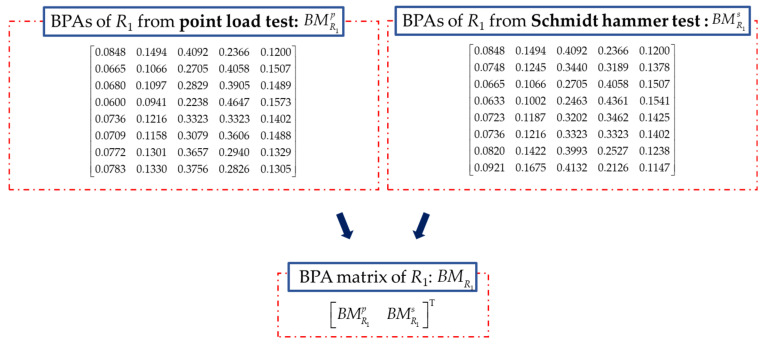
Constructed with BPAs of *R*_1_.

**Figure 8 sensors-21-07108-f008:**
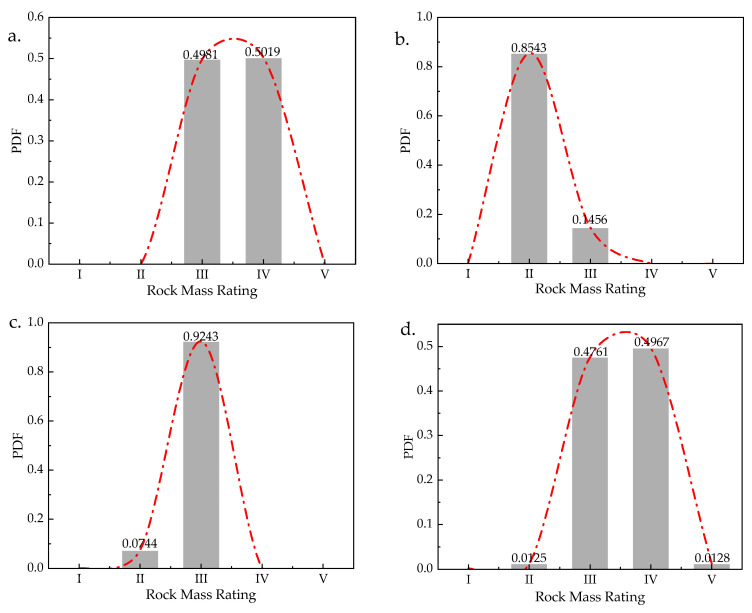
Initial fusion results of each rock mass index (*R*_1_–*R*_5_). (**a**) Initial fusion result of *R*_1_. (**b**) Initial fusion result of *R*_2_. (**c**) Initial fusion result of *R*_3_. (**d**) Initial fusion result of *R*_4_. (**e**) Initial fusion result of *R*_5_.

**Figure 9 sensors-21-07108-f009:**
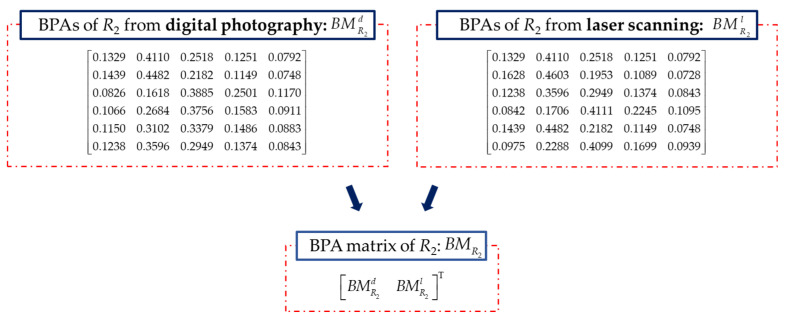
$BM_{R_{2}}$ constructed with BPAs of *R*_2_.

**Figure 10 sensors-21-07108-f010:**
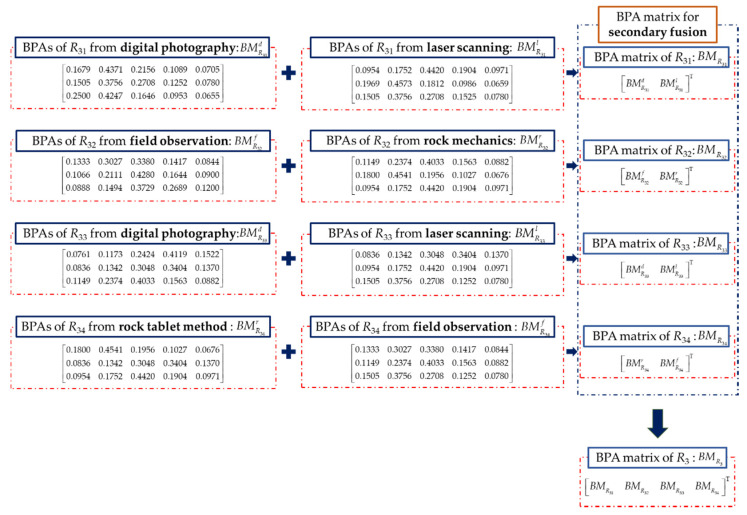
$BM_{R_{3}}$ constructed with BPAs of *R*_3_.

**Figure 11 sensors-21-07108-f011:**
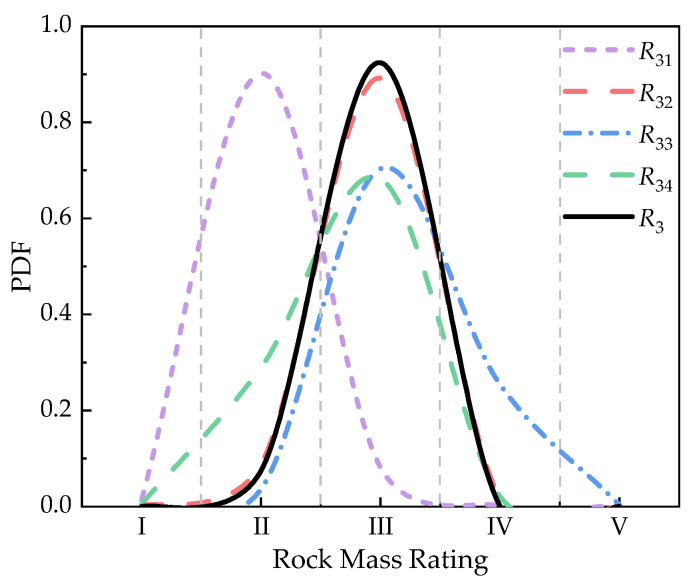
Result of the secondary fusion of the sub-indicators of *R*_3_.

**Figure 12 sensors-21-07108-f012:**
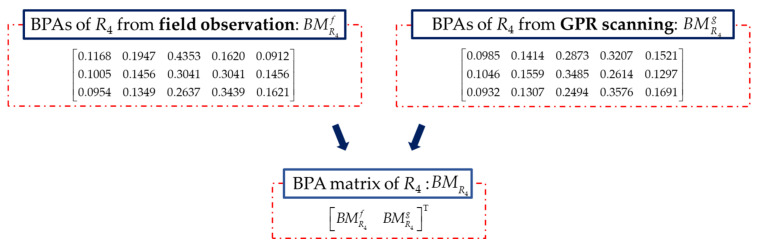
$BM_{R_{4}}$ constructed with BPAs of *R*_4._

**Figure 13 sensors-21-07108-f013:**
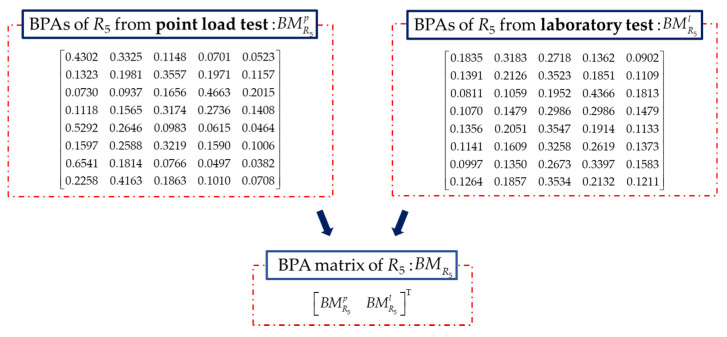
$BM_{R_{5}}$ constructed with BPAs of *R*_5_.

**Figure 14 sensors-21-07108-f014:**
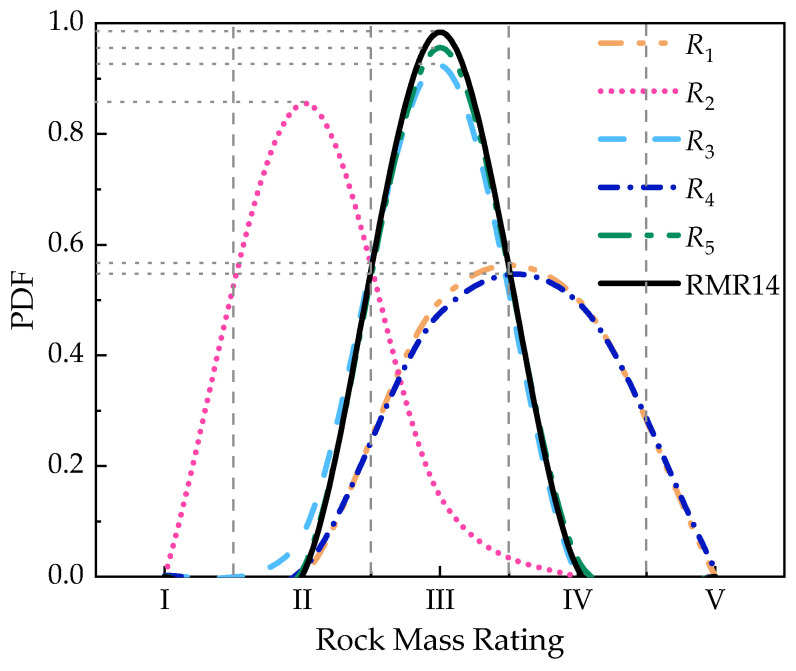
Fusion result of the initial data fusion in each index and final data fusion in RMR14.

**Table 1 sensors-21-07108-t001:** Standardized ranges for rock mass rating indices.

Rating Index	I	II	III	IV	V
Rock strength (*R*_1_)	12–15	8–12	5–8	3–5	0–3
Spacing of discontinuities (*R*_2_)	29–40	22–29	15–22	8–15	0–8
Condition of discontinuities (*R*_3_)	17–20	13–17	9–13	4–9	0–4
Ground water (*R*_4_)	13–15	10–13	7–10	3–7	0–3
Intact rock alterability (*R*_5_)	9–10	7–9	4–7	2–4	0–2
Total rating	80–100	60–80	40–60	20–40	0–20

**Table 2 sensors-21-07108-t002:** Rating ranges of sub-indicators of *R*_3_.

Rock Mass Rating	I	II	III	IV	V
Continuity (*R*_31_)	4.1–5.0	3.1–4.0	2.1–3.0	1.1–2.0	0–1.0
Roughness (*R*_32_)	4.1–5.0	3.1–4.0	2.1–3.0	1.1–2.0	0–1.0
Infilling (*R*_33_)	4.1–5.0	3.1–4.0	2.1–3.0	1.1–2.0	0–1.0
Weathering (*R*_34_)	4.1–5.0	3.1–4.0	2.1–3.0	1.1–2.0	0–1.0

**Table 3 sensors-21-07108-t003:** Data of rock mass for model validation.

No.	*R* _1_	*R* _2_	*R* _3_	*R* _4_	*R* _5_
1	14	11	12	3	2
2	13	30	12	4	10
3	5	20	17	13	7
4	10	19	16	10	5
5	3	36	12	7	5

**Table 4 sensors-21-07108-t004:** Fusion process of the model validation.

	Rock Mass Rating				
	I	II	III	IV	V
$\tilde{RB}$	0.4397	0.4720	0.3054	0.2494	0.2401
*Se*	2.2744	2.1187	3.2741	4.0091	4.1646
*c*	0.1436	0.1338	0.2067	0.2531	0.2629

**Table 5 sensors-21-07108-t005:** Fusion process of the quality evaluation of rock mass in studied section.

	Rock Mass Rating				
	I	II	III	IV	V
$\tilde{RB}$	0.4819	0.7590	0.4779	0.4751	0.4625
*Se*	2.0749	1.3176	2.0927	2.1046	2.1622
*c*	0.2128	0.1351	0.2146	0.2158	0.2217
Weighted average belief	6.2 × 10^−4^	0.1366	**0.6387**	0.2213	0.0028
Final probability	8.7 × 10^−13^	0.0021	**0.9838**	0.0142	3.5 × 10^−10^

## Data Availability

The data presented in this study are available on request from the corresponding author.

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
