# Peer review of "Quality Evaluation of Rock Mass Using RMR14 Based on Multi-Source Data Fusion"

_sensors, 2021, doi:10.3390/s21217108_

Round 1
Reviewer 1 Report
This paper proposed a multi-source quality evaluation method based on D-S evidence theory for rock mass. Technical contributions and writing formats of this paper should be further improved. The following comments are provided.
1) The Introduction is not well prepared. The authors should give a brief illustration of the basic principles and advantages of this paper. In addition, the novelty should be emphasized in Introduction.
2) Comparative results are not enough. More details should be provided. Other better algorithms should be considered in comparatives.
3) By considering the topic presented in this paper, the authors should give some future directions in Conclusion.
4) The authors should give the reasonableness for setting the evaluation criteria.
5) For complex systems like rock mass, time-consuming is a critical problem. How is the time-consuming of the proposed method? Does this affect its applications?
6) Discussions should be added for discussing the pros and cons of the proposed method.
Author Response
This paper proposed a multi-source quality evaluation method based on D-S evidence theory for rock mass. Technical contributions and writing formats of this paper should be further improved. The following comments are provided.
Thank you for the constructive and critical comments concerning our manuscript which are very helpful in restructuring and improving the paper. Detailed responses are listed below. Main changes are marked in RED in the revised manuscript.
1. The Introduction is not well prepared. The authors should give a brief illustration of the basic principles and advantages of this paper. In addition, the novelty should be emphasized in Introduction.
Reply: The introduction has been revised. The principles and advantages of this paper are mentioned in the introduction section, as well as the novelty.
2. Comparative results are not enough. More details should be provided. Other better algorithms should be considered in comparatives.
Reply: We have compared our method to the method based on the traditional D-S evidence theory. Now we also compare our method to the traditional evaluation method in RMR14 system, which is most widely used nowadays. The comparison is introduced more detailed in Section 3.3.
3. By considering the topic presented in this paper, the authors should give some future directions in Conclusion.
Reply: Some further directions of this paper are put forward in the last paragraph of the conclusion section to improve the proposed method.
4. The authors should give the reasonableness for setting the evaluation criteria.
Reply: The setting of the evaluation criteria is based on RMR14 system, which is one of the relatively perfect systems for the evaluation of rock mass in tunnel engineering project. RMR14 system and the setting of the evaluation criteria were proposed by Celada et al. (2014).
References:
Celada, B.; Tardáguila, I.; Varona, P.; Rodríguez, A.; Bieniawski, Z.T. Innovating Tunnel Design by an Improved Experience-Based RMR System. Proceedings of the World Tunnel Congress. Foz do Iguaçu, Brazil; 2014; pp. 1–9.
5. For complex systems like rock mass, time-consuming is a critical problem. How is the time-consuming of the proposed method? Does this affect its applications?
Reply: In its application, the proposed method does not actually take much time. Based on the improvement and optimization of the algorithm, when there is a large amount of data of rock mass input, the proposed method still can output the results of the quality evaluation of rock mass in a few minutes. It is also one of the advantages of this method.
6. Discussions should be added for discussing the pros and cons of the proposed method.
Reply: The discussions of the pros and cons of the proposed method have been added in the conclusion section. As for the more reliable application of this method, we also put forward some directions for the future research of this paper.
Reviewer 2 Report
The topic Quality Evaluation of Rock Mass Using RMR14 Based on Multi-source Data Fusion potentially interesting, however, there are some issues that should be addressed by the authors:
- The Introduction" sections can be made much more impressive by highlighting your contributions. The contribution of the study should be explained simply and clearly.
- The authors should further enlarge the Introduction with current work about machine learning-based multi-data fusion methods to improve the research background, for example: Fusion of vibration and current signatures for the fault diagnosis of induction machines; Effective multi-sensor data fusion for chatter detection in milling process.
- The reviewer recommends a comprehensive English review. There are some minor issues.
- Figure 14 should be revised to make more clear (R1 and R2 curves)
- Conclusion section should be rearranged. According to the topic of the paper, the authors may propose some interesting problems as future work in the conclusion.
This study may be proposed for publication if it is addressed in the specified problems.
Author Response
The topic Quality Evaluation of Rock Mass Using RMR14 Based on Multi-source Data Fusion potentially interesting, however, there are some issues that should be addressed by the authors. This study may be proposed for publication if it is addressed in the specified problems.
Thank you for the constructive and critical comments concerning our manuscript which are very helpful in restructuring and improving the paper. Detailed responses are listed below. Main changes are marked in RED in the revised manuscript.
1. The Introduction" sections can be made much more impressive by highlighting your contributions. The contribution of the study should be explained simply and clearly.
Reply: Our contributions of the study have been briefly explained and added in the section of introduction.
2. The authors should further enlarge the Introduction with current work about machine learning-based multi-data fusion methods to improve the research background, for example: Fusion of vibration and current signatures for the fault diagnosis of induction machines; Effective multi-sensor data fusion for chatter detection in milling process.
Reply: The introduction with current work about machine learning based on multi-source data fusion has been added in the section of introduction and some related references are cited.
3. The reviewer recommends a comprehensive English review. There are some minor issues.
Reply: Thank you for your advice. We have invited an English native speaker to improve our writing in the paper and fixed some minor issues.
4. Figure 14 should be revised to make more clear (R1 and R2 curves)
Reply: Figure 14 has been revised to a clearer version. Color and type of the lines are changed for it easier to be observed.
5. Conclusion section should be rearranged. According to the topic of the paper, the authors may propose some interesting problems as future work in the conclusion.
Reply: Conclusions have been revised. The structure of the conclusion section is rearranged, and some possible directions for the further research of this paper are put forward in the last paragraph of the conclusion section.
Round 2
Reviewer 1 Report
The authors have addressed the problems that the reviewer considered before. Now I have some minor concerns.
1) Please enhance the Introduction by, for example, considering data-driven fault diagnosis for traction systems in high-speed trains: A survey, challenges, and perspectives.
2) Some open issues and limitations of the proposed methods should be added to discussions.
3) The authors should add a summary of the current studies in the Introduction.
Author Response
The authors have addressed the problems that the reviewer considered before. Now I have some minor concerns.
Thank you for the constructive and critical comments concerning our manuscript which are very helpful in restructuring and improving the paper. Detailed responses are listed below. Main changes are marked using the “Track Changes” function in the revised manuscript.
1. Please enhance the Introduction by, for example, considering data-driven fault diagnosis for traction systems in high-speed trains: A survey, challenges, and perspectives.
Reply: The introduction with current study about data-driven computing has been added in the Introduction and the related reference is cited.
2. Some open issues and limitations of the proposed methods should be added to discussions.
Reply: The limitations of our proposed method have been discussed in Conclusion. There are two major limitations, including without consideration the correlations of the rating indexes and the applicability of the proposed method on other rock mass engineering projects.
3. The authors should add a summary of the current studies in the Introduction.
Reply: The summary of the current studies has been added in the Introduction, including the summary of the current study on multi-source data fusion. The summary of the features of D-S evidence theory was in the first paragraph of page 3 already.
Reviewer 2 Report
The authors have improved the quality of the paper. I have no further question.
Author Response
The authors have improved the quality of the paper. I have no further question.
Thank you again for the constructive and critical comments concerning our manuscript.